# Learning to Predict Vehicle Trajectories with Model-based Planning

**Haoran Song   Di Luan   Wenchao Ding   Michael Yu Wang   Qifeng Chen**
The Hong Kong University of Science and Technology
`{hsongad, wdingae, dluan, mywang, cqf}@ust.hk`

**Abstract:** Predicting the future trajectories of on-road vehicles is critical for autonomous driving. In this paper, we introduce a novel prediction framework called PRIME, which stands for Prediction with Model-based Planning. Unlike recent prediction works that utilize neural networks to model scene context and produce unconstrained trajectories, PRIME is designed to generate accurate and feasibility-guaranteed future trajectory predictions. PRIME guarantees the trajectory feasibility by exploiting a model-based generator to produce future trajectories under explicit constraints and enables accurate multimodal prediction by utilizing a learning-based evaluator to select future trajectories. We conduct experiments on the large-scale Argoverse Motion Forecasting Benchmark, where PRIME outperforms the state-of-the-art methods in prediction accuracy, feasibility, and robustness under imperfect tracking.

**Keywords:** trajectory prediction, autonomous driving

## 1   Introduction

In the architecture of autonomous driving, prediction serves as the bridging module that reasons future states based on the perceived information from upstream detection and tracking and provides the predicted future states to facilitate the downstream planning. Therefore, making accurate and reasonable trajectory predictions for on-road vehicles is vital for planning safe, efficient, and comfortable motion for self-driving vehicles (SDVs).

The widely known challenge of trajectory prediction lies in modeling multi-agent interaction and inferring multimodal future states under driving scenarios. Traditional methods [1, 2, 3, 4, 5] produce motion forecasting by handcrafted rules or models with embedded physical and environmental features, which are insufficient for modeling interactive agents in complex scenes. Learning-based approaches [6, 7, 8], with deep neural networks to fuse scene context information and generate future trajectories, significantly promote the prediction accuracy and dominate the recent motion forecasting benchmarks for autonomous driving [9, 10].

Despite achieving steady improvement in accuracy, much less attention has been paid to the feasibility and robustness of learning-based trajectory prediction. Indeed, most traffic participants operate under their inherent kinematic constraints (e.g., non-holonomic motion constraints for vehicles) while in compliance with the road structure (e.g., lane connectivity, static obstacles) and semantic information (e.g., traffic lights, speed limits). All these kinematic and environmental constraints explicitly regularize the trajectory space. However, most existing approaches model traffic agents as points and generate future trajectories without imposing constraints. Such constraint-free predictions may be incompliant with kinematic or environmental characteristics and thus give rise to massive uncertainty in the predicted future states. As a result, the downstream planning module would inevitably undergo some extra burdens, and even the "freezing robot problem" [11]. Furthermore, the trajectory predictions typically generated by neural network regression have high dependences on long-term tracking. For some dense driving scenarios where the target would be momently occluded or suddenly appears within the sensing range, tracking results are discontinuous or not accumulated enough. The prediction accuracy would thereby degrade under such imperfect tracking cases.

5th Conference on Robot Learning (CoRL 2021), London, UK.

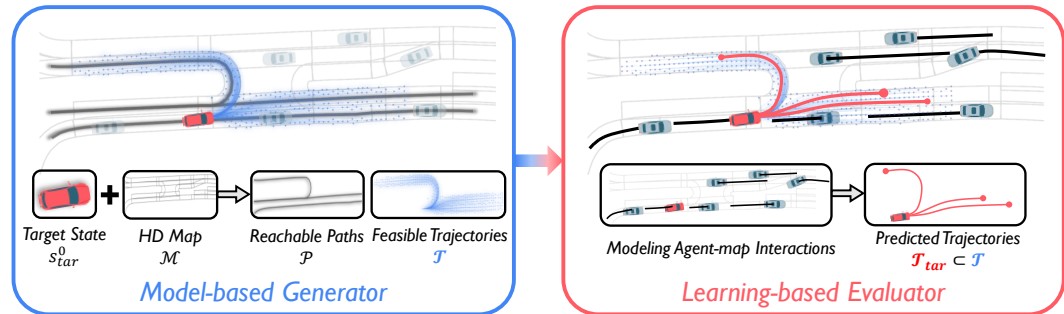

Figure 1: Illustration of the PRIME framework. The model-based generator (left) samples feasible future trajectories $\mathcal{T}$ for the target agent by taking its real-time state $\mathbf{s}_{tar}^0$ and HD map $\mathcal{M}$, while imposing explicit constraints $\mathcal{C}$ to guarantee trajectory feasibility. The learning-based evaluator (right) receives the feasible trajectory set $\mathcal{T}$ and all observed tracks $\mathcal{S}$ to model the implicit interactions in scene context and then selects a final set of trajectories $\mathcal{T}_{tar} \subset \mathcal{T}$ as the prediction result.

Toward overcoming these challenges, we propose PRIME, a novel architecture for vehicle trajectory prediction, as illustrated in Fig. 1. The core idea is to exploit a model-based motion planner as the prediction generator to produce feasibility-guaranteed future trajectories under explicit physical constraints, together with a deep neural network as the prediction evaluator to enable accurate multimodal prediction by learning complex implicit interactions. To the best of our knowledge, PRIME is the first to incorporate an interpretable motion planner in prediction learning and also the only method that ensures kinematic and environmental feasibility in data-driven trajectory prediction. We conduct experiments on the large-scale Argoverse motion forecasting benchmark and achieves better prediction accuracy over the state-of-the-art. Furthermore, PRIME shows significant superiority in trajectory feasibility guarantee and prediction robustness under imperfect tracking. These attributes would facilitate more flexible and safe motion planning for SDVs.

## 2   Related Work

**Prediction and Planning** are closely intertwined in autonomous driving [12, 13, 14, 15]. Planning is to generate constraint-compliant trajectory candidates and, after considering safety, comfort, path progress, etc., select the best trajectory for execution by the SDV (ego agent). Prediction facilitates the trajectory selection in planning by inferring future trajectories of the surrounding vehicles (target agents). Their different focuses make the corresponding mainstream solutions diverge. Model-based approaches [16, 17, 18, 19] are preferred in planning due to their interpretability and reliability in computing safe trajectories under explicit constraints. Learning-based methods [6, 7, 20, 21], in contrast, prevail in prediction by utilizing its advantage in modeling implicit interactions.

Some learning-based prediction works incorporate the goal-directed idea from planning to infer the possible goals and then produce goal-conditioned trajectories [22, 23, 24, 25]. Moreover, the novel planning-prediction-coupled frameworks are introduced to make predictions conditioned on ego intentions [26] or motion plans [27, 28]. Although much emphasis on improving the point-level prediction accuracy, the data-driven frameworks cannot ensure the given constraints are indeed imposed on trajectory generation. Despite DKM [29] embeds the two-axle vehicle kinematics [30] in the output layer to ensure kinematic feasibility, yet still no guarantee on environmental compliance. Inspired by the popular sampling-based paradigm in vehicle motion planning [13, 17], we employ a model-based planner for providing feasibility-guaranteed trajectory sets, and thereby the learning-based part is reduced to evaluate future trajectories. With making the most of model-based planning and learning-based prediction, PRIME handles complex agent-map interactions while fulfilling environmental and kinematic constraints.

**Modeling agent-map interactions** is fundamental for capturing information from scene map and dynamic agents. The rasterized representation [20, 31, 32] is proposed for learning-based methods, which renders traffic entities into images by different colors or intensities and then encodes rasters with Convolutional Neural Networks. As an alternative, the vectorized representation [33, 34, 25] vectorizes scene context as nodes to construct a graph, which exploits High Definition (HD) maps more explicitly and improves prediction accuracy. By contrast, we address the agent-map modeling

with a hierarchical structure that incorporates the lane-association ideas from [3] while extends to learn global scene context. To be specific, our prediction generator acts locally in a planning manner to generate path-conditioned trajectory sets, and the prediction evaluator learns a global understanding of the scene context by aggregating all trajectory and map features.

**Generating multimodal trajectories** is essential for handling the intrinsic multimodal prediction distributions. Stochastic models are mostly built upon conditional variational autoencoder [7, 35, 36, 37, 38] and generative adversarial network [39, 40, 41, 42], while sampling with uncontrollable latent variables at inference may impede their deployment on safety-critical driving systems. Deterministic models are mainly based on multi-mode trajectory regression [43, 44, 20, 34]. To alleviate mode collapse in prediction learning, recent works decompose the task into classification over anchor trajectories [45] or goal-conditioned trajectories [25], followed by trajectory offset regression. However, no feasibility could be ensured for the regressed results. CoverNet [32] formulates multimodal prediction by directly classifying on a pre-constructed trajectory set, but still, its predictions may violate the agent kinematics or scene constraints. For our framework, leveraging model-based planning as the prediction generator brings superiorities in 1) maintaining multimodal distributions by generating trajectories on diverse reachable paths, 2) ensuring trajectory feasibility by imposing real-time constraints, 3) mitigating the high reliance on long-term tracking, and 4) producing trajectories with continuous information rather than just discrete locations.

# 3 Overview

**Problem formulation.** Assume the self-driving vehicle is equipped with detection and tracking modules to provide observed states $\mathcal{S}$ of on-road agents $\mathcal{A}$ and has access to HD map $\mathcal{M}$. Let $\mathbf{s}_i^t$ denote the state of agent $a_i \in \mathcal{A}$ at frame $t$, including position, heading, velocity, turning rate and actor type, and $\mathbf{s}_i = \left\{ \mathbf{s}_i^{-T_P+1}, \mathbf{s}_i^{-T_P+2}, ..., \mathbf{s}_i^0 \right\}$ denotes the state sequence in the observed period $T_P$. Given any agent as the prediction target, we denote it by $a_{tar}$ and its surrounding agents by $\mathcal{A}_{nbrs} = \{a_1, a_2, ..., a_m\}$ for differentiation, with their state sequence correspondingly given as $\mathbf{s}_{tar}$ and $\mathcal{S}_{nbrs} = \{\mathbf{s}_1, \mathbf{s}_2, ..., \mathbf{s}_m\}$. Accordingly, $\mathcal{S} = \{\mathbf{s}_{tar}\} \cup \mathcal{S}_{nbrs}$ and $\mathcal{A} = \{a_{tar}\} \cup \mathcal{A}_{nbrs}$. Our objective is to predict multimodal future trajectories $\mathcal{T}_{tar} = \{\mathcal{T}_k | k = 1, 2, ..., K\}$ together with corresponding trajectory probability $\{p_k\}$, where $\mathcal{T}_k$ denotes a predicted trajectory for target agent $a_{tar}$ with continuous state information up to the prediction horizon $T_F$, $K$ is the number of predicted trajectories. Additionally, it is required to ensure each prediction $\mathcal{T}_k \in \mathcal{T}_{tar}$ is feasible with existing constraints $\mathcal{C}$, which includes environmental constraints $\mathcal{C}_{\mathcal{M}}$ and the kinematic constraints $\mathcal{C}_{tar}$.

**Our framework.** The two-stage architecture of PRIME consists of model-based generator $G$ and learning-based evaluator $E$. Concretely, the generator $G : (\mathbf{s}_{tar}^0, \mathcal{M}, \mathcal{C}) \mapsto (\mathcal{P}, \mathcal{T})$ is tasked to produce the trajectory space for the target, which is approximated by a finite set of feasible trajectories $\mathcal{T}$. This part starts with searching a set of reachable paths $\mathcal{P} = \{\mathcal{P}_j | j = 1, 2, ..., l\}$ from HD map $\mathcal{M}$, which provides reference path for trajectory generation. Then a classical sampling-based planner is utilized to generate trajectory samples under constraints in $\mathcal{C}$, and thus provide the feasible future trajectory set $\mathcal{T} = \bigcup_{j=1}^l \{\mathcal{T}_{j,k} | k = 1, 2, ..., n_j\}$ for the target. $\mathcal{T}_{j,k}$ denotes the $k$-th feasible trajectory generated from path $\mathcal{P}_j$, and the total number of trajectories is $n = \sum_{j=1}^l n_j$. The model-based part is specialized in generating trajectories with feasibility guaranteed but ignores multi-agent interactions. The evaluator $E : (\mathcal{P}, \mathcal{T}, \mathcal{S}) \mapsto (\mathcal{T}_{tar}, \{p_k\})$ takes charge of learning implicit interactions, which features with a dual representation for spatial information and with the attention mechanism to process the varying sizes of $l$ reachable paths, $m$ surrounding agents, and $n$ feasible trajectories. Notably, the evaluator $E$ is reduced to score trajectories and select prediction results $\mathcal{T}_{tar} \subset \mathcal{T}$, rather than regressing position or displacement as most learning-based frameworks do.

# 4 Model-based Generator

## 4.1 Path Search

Unlike motion planning, where the reference path for the controllable ego agent is given, the future paths of uncontrollable targets in prediction are unknown. Therefore, we conduct the path search in advance of trajectory generation such that any prediction target could be associated with a set of potential paths $\mathcal{P}^+$. Our path search algorithm $G_{path} : (\mathcal{M}, \mathbf{s}_{tar}^0) \mapsto \mathcal{P}^+$ is implemented by Depth-

First-Search on HD map $\mathcal{M}$, with more details described in the supplementary material. Yielding a potential path $\mathcal{P}_j \in \mathcal{P}^+$ with the centerline coordinates of each lane segment sequence, we expect all the paths of $\mathcal{P}^+$ to provide sufficient coverage for the future trajectory space of $a_{tar}$. As no dynamic constraint is imposed in this phase, for target with current state $\mathbf{s}_{tar}^0$, some paths in $\mathcal{P}^+$ may not be reachable at frame $t = T_F$. For instance, a high-speed vehicle cannot change to the opposite lane with a U-turn in few seconds. Such unreachable paths could be recognized in the following trajectory generation phase as no trajectories samples towards them are feasible. Finally, a set of reachable paths $\mathcal{P} \subseteq \mathcal{P}^+$ would be reserved.

## 4.2 Trajectory Generation

Given the potential paths in $\mathcal{P}^+$ as dynamic references, we choose to generate future trajectories in a planning manner. For SDV, motion planning typically aims at finding an optimal trajectory to connect the current state and a goal state, essentially different from prediction that infers probable trajectories for vehicles with unknown intentions. Despite this, the model-based generator in planning, which computes a large number of trajectory samples for the follow-up selection, could also be exploited in prediction.

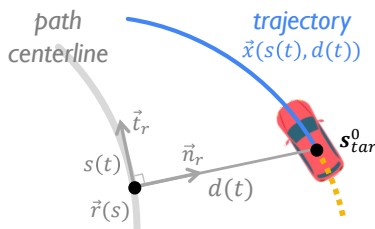

Figure 2: Trajectory generation in a Frenét Frame

We adopt the trajectory generation phase of Frenét planner [17] in our trajectory generator $G_{traj} : (\mathcal{P}^+, \mathbf{s}_{tar}^0, \mathcal{C}) \mapsto \mathcal{T}$. Given a reference path in $\mathcal{P}^+$, a dynamic curvilinear frame is given by the tangential vector $\vec{t}_r$ and normal vector $\vec{n}_r$ at a certain point $r$ on the path centerline. The Cartesian coordinate $\vec{x} = (x, y)$ could be converted to the Frenét coordinate $(s, d)$, with the relation

$$\vec{x}(s(t), d(t)) = \vec{r}(s(t)) + d(t)\vec{n}_r(s(t)), \tag{1}$$

in which $\vec{r}$ represents a vector pointing from the path root, $s$ and $d$ denote the covered arc length and the perpendicular offset, as illustrated in Fig. 2. The trajectory generation phase first projects the current state $\mathbf{s}_{tar}^0$ onto the Frenét frame and obtains the state tuple $[s_0, \dot{s}_0, \ddot{s}_0, d_0, \dot{d}_0, \ddot{d}_0]$. The longitudinal movement $s(t)$ and lateral movement $d(t)$ within the prediction horizon $T_F$ are then generated independently by connecting the fixed start state with different end states using parametric curves to cover different driving maneuvers. Compared with planning, prediction receives less accurate state estimation for targets and does not need fine-grained trajectories. In our trajectory generator, therefore, some high-order state variables are simplified to zero. For longitudinal movement, we sample the target velocity $\dot{s}(T_F) \leftarrow \dot{s}_i$ in the range of $[\max(0, \dot{s}_0 - \delta^- T_F), \min(\hat{s}, \dot{s}_0 + \delta^+ T_F)]$ while leaving $s(T_F)$ unconstrained. The constants $\delta^-$, $\delta^+$ and $\hat{s}$ are given by considering the actor type of $a_{tar}$ and speed limit in $\mathcal{M}$, to control the longitudinal velocity $\dot{s}_i$ in a reasonable range. Each longitudinal trajectory $s_i(t)$ is calculated using a quartic polynomial

$$\text{s.t.} \quad [s(0), \dot{s}(0), \ddot{s}(0), \dot{s}(T_F), \ddot{s}(T_F)] = [s_0, \dot{s}_0, 0, \dot{s}_i, 0].$$

For lateral movement, we sample the target offset $d(T_F) \leftarrow d_j$ in the range of $[-d_{lane}/2, d_{lane}/2]$, where $d_{lane}$ denotes lane width. Each lateral trajectory $d_j(t)$ is calculated using a quintic polynomial

$$\text{s.t.} \quad [d(0), \dot{d}(0), \ddot{d}(0), d(T_F), \dot{d}(T_F), \ddot{d}(T_F)] = [d_0, \dot{d}_0, 0, d_j, 0, 0].$$

With the resulted longitudinal and lateral trajectory set $\mathcal{T}_{lon}$ and $\mathcal{T}_{lat}$, a full trajectory $\vec{x}(s(t), d(t))$ is formed by every combinations in $\mathcal{T}_{lon} \times \mathcal{T}_{lat}$. Next, the trajectories incompliant with given constraints $\mathcal{C}$ would be filtered out. We first project the Frenét coordinates $(s, d)$ back to global coordinates $(x, y)$ to check if the trajectory collides with static obstacles given in $\mathcal{C}_{\mathcal{M}}$. For collision-free trajectories, their high-order state variables are then converted by the Frenét-Cartesian-transfomation

$$[s, \dot{s}, \ddot{s}, d, \dot{d}, \ddot{d}] \longmapsto [\vec{x}, v, \kappa, \alpha] \tag{2}$$

to check if any velocity $v$, acceleration $\alpha$ or curvature $\kappa$ exceeds the kinematic constraints given in $\mathcal{C}_{tar}$. Finally, each reference path $\mathcal{P}_j \in \mathcal{P}$ would generate a set of $n_j$ feasibility-guaranteed future trajectories $\{\mathcal{T}_{j,k} | k = 1, 2, ..., n_j\}$, and all the trajectories together form an overall trajectory space $\mathcal{T}$. Although the constraints are set conservatively with leaving some margin for the learning-based evaluator, our model-based generator effectively narrows down the trajectory space $\mathcal{T}$ by imposing constraints. This unique advantage would set our framework with higher accuracy and robustness.

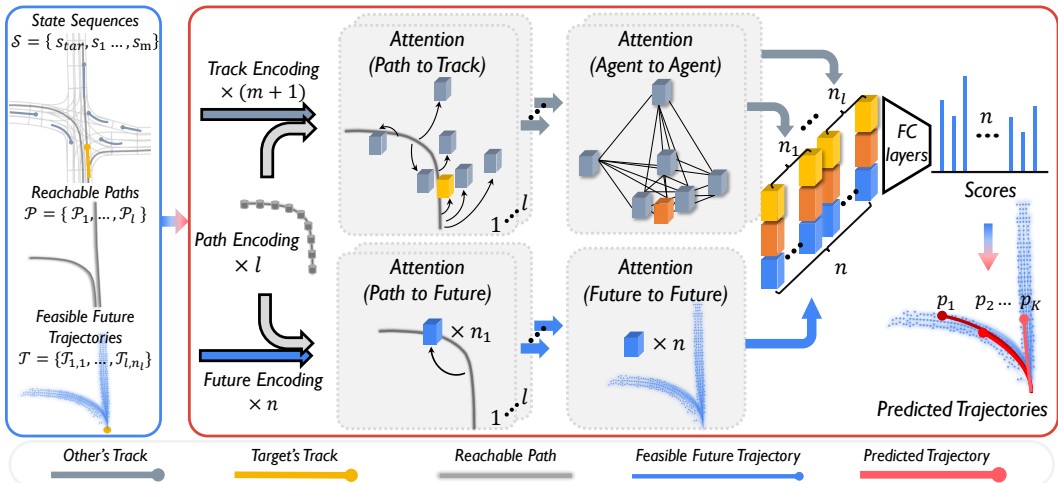

Figure 3: PRIME framework overview. The model-based generator searches reachable paths $\mathcal{P}$ through the map and produces feasible future trajectories $\mathcal{T}$. The learning-based evaluator encodes the traffic entities in $(\mathcal{P}, \mathcal{T}, \mathcal{S})$ and learns implicit interactions in the subsequent attention modules. Afterwards, each future trajectory $\mathcal{T}_{j,k}$ could query its track tensor $\mathbf{X}_j(\mathbf{s}_{tar})$ from P2T, interaction tensor $\mathbf{Y}_j(\mathbf{s}_{tar})$ from A2A and future tensor $\mathbf{Z}(\mathcal{T}_{j,k})$ from F2F, and it is scored by feeding the concatenation of these tensors to fully-connected layers. Finally, the evaluator ranks all feasible future trajectories in $\mathcal{T}$ by scoring and outputs a final set of $K$ predicted trajectories.

## 5 Learning-based Evaluator

### 5.1 State Representation

The prediction evaluator aggregates scene context, including observed state sequences $\mathcal{S}$, path set $\mathcal{P}$, and future trajectory set $\mathcal{T}$. To make it compatible with most existing trajectory prediction datasets, state sequence $\mathbf{s}_i$ is reduced to history track in the learning part. Before feeding to the network, we discretize each history track $\mathbf{s}_i$ and future trajectory $\mathcal{T}_{j,k}$ as a location sequence with time interval $\Delta T$, and each reference path $\mathcal{P}_j$ as a waypoint sequence with distance interval $\Delta D$. Since the longitudinal movement $s$ and lateral offset $d$ indicate how an agent moves relative to a reference path, they represent the local spatial relationship more straightforwardly. For this reason, we use the Frenét coordinates $(s, d)$ in addition to the Cartesian coordinates $(x, y)$ to form a dual spatial representation. Here, the spatial information $(x, y, s, d)$ of future trajectories in $\mathcal{T}$ is given by the generator, while the $(s, d)$ coordinates of history tracks in $\mathcal{S}$ are obtained by projecting $(x, y)$ coordinates on the corresponding reference path. Additionally, we adopt the approach of [34] to add a binary mask $b$ to history track's representation $(x, y, s, d, b)$ to indicate if the location is padded.

### 5.2 Encoding Scene Context

Prior to capture interrelationships between traffic entities, we first encode each kind of entity in the scene. All encoders are structured with a temporal convolutional layer followed by a long short-term memory (LSTM) layer. The track encoder and the future encoder employ a unidirectional LSTM and make the last hidden state $h(\cdot)$ as the motion encoding for history track and future trajectory, while the path encoder uses a bidirectional LSTM and provides the sequence of hidden states $H(\cdot)$ as the path spatial encoding. Given the scene context description $(\mathcal{S}, \mathcal{P}, \mathcal{T})$, each reachable path $\mathcal{P}_j \in \mathcal{P}$ is encoded as a $H(\mathcal{P}_j)$, where $j = 1, 2, ..., l$. Considering that the Frenét representation is dependent with the path frame, we encode all history tracks with respect to each reference path $\mathcal{P}_j$, which results in $l$ groups of track encodings $\{h(\mathbf{s}_{tar}), h(\mathbf{s}_1), ..., h(\mathbf{s}_m)\}_j$. Each future trajectory $\mathcal{T}_{j,k} \in \mathcal{T}$ is relative to its reference path $\mathcal{P}_j$, so all future trajectories are encoded correspondingly to form $l$ groups of future encodings $\{h(\mathcal{T}_{j,k})|k = 1, 2, ..., n_j\}$.

### 5.3 Modeling Interactions

Next is to capture the implicit interactions resulted from the static environment and multiple dynamic agents. To fuse the spatial-temporal information from varying numbers of entities in the scene con-

text, the attention mechanism [46] is adopted to construct four modules, namely, path to track (P2T), path to future (P2F), agent to agent (A2A), and future to future (F2F). They are implemented in the same way of scaled dot-product attention and use linear layers for mapping key, query and value. The overall workflow is shown in Fig. 3. In the upper branch, P2T brings the spatial information of each path encoding $\mathcal{P}_j$ into the corresponding track encodings $\{h(\mathbf{s}_{tar}), h(\mathbf{s}_1), ..., h(\mathbf{s}_m)\}_j$. The track encodings are further processed by a self-attention structure in A2A, aiming to capture the interactions between agents in the past time domain. The lower branch lays emphasis on updating the features contained in future encodings. P2F brings the spatial information of path encoding $H(\mathcal{P}_j)$ into the corresponding future encodings $\{h(\mathcal{T}_{j,k})|k = 1, 2, ..., n_j\}$. It is followed by F2F that fuses all future encodings $\bigcup_{j=1}^{l} \{h(\mathcal{T}_{j,k})|k = 1, 2, ..., n_j\}$ from different paths $\mathcal{P}_j(j = 1, 2, ..., l)$ using self-attention. In particular, F2F obtains a global understanding of the reachable space given by $\mathcal{P}$ and, by this way, attempts to further perceive the differences between different trajectories in $\mathcal{T}$. For any future trajectories $\mathcal{T}_{j,k} \in \mathcal{T}$, the corresponding track tensor $\mathbf{X}_j(\mathbf{s}_{tar})$, interaction tensor $\mathbf{Y}_j(\mathbf{s}_{tar})$ and future tensor $\mathbf{Z}(\mathcal{T}_{j,k})$ could be obtained from P2T, A2A and F2F modules, which are then concatenated together to form a full description $\mathbf{U}_{j,k} = \text{Concat}(\mathbf{X}_j(\mathbf{s}_{tar}), \mathbf{Y}_j(\mathbf{s}_{tar}), \mathbf{Z}(\mathcal{T}_{j,k}))$.

### 5.4 Trajectory Scoring, Learning, and Inference

With $\mathbf{U}_{j,k}$ as a full description, we score all the $n$ trajectories $\mathcal{T}_{j,k}$ using a maximum entropy model:

$$\gamma(\mathcal{T}_{j,k}) = \frac{\exp(f(\mathbf{U}_{j,k}))}{\sum_{j=1}^{l} \sum_{k=1}^{n_j} \exp(f(\mathbf{U}_{j,k}))}, \tag{3}$$

in which $f(\cdot)$ is implemented using a 3-layer MLP at the end of the evaluation network $E$. The score label $\psi(\mathcal{T}_{j,k})$ is resulted from calculating the accumulated squared distance error $\text{Dist}(\cdot)$ between the future trajectory $\mathcal{T}_{j,k}$ and the ground truth trajectory $\mathcal{T}_{GT}$ within the prediction horizon $T_F$:

$$\psi(\mathcal{T}_{j,k}) = \frac{\exp(-\text{Dist}(\mathbf{T}_{j,k}, \mathbf{T}_{GT})/\tau)}{\sum_{j=1}^{l} \sum_{k=1}^{n_j} \exp(-\text{Dist}(\mathbf{T}_{j,k}, \mathbf{T}_{GT})/\tau)}, \tag{4}$$

where $\tau$ is set as a temperature factor. The overall network is trained by cross entropy between the evaluated scores and the labeled scores $\mathcal{L} = \text{CrossEntropy}(\gamma(\mathcal{T}_{j,k}), \psi(\mathcal{T}_{j,k}))$. For the inference stage that requires $K$ predicted trajectories, we adopt the non-maximum suppression (NMS) algorithm to remove near-duplicate trajectories, as did in [25]. According to the predicted scores, this method greedily picks trajectories from $\mathcal{T}$ and excludes the lower scored trajectory between very close ones. Finally, $K$ trajectories with descending order of scores form the prediction result $\mathcal{T}_{tar} = \{\mathcal{T}_i|k = 1, 2, ..., K\}$, and the prediction probability $p_k$ is derived by $p_k = \gamma(\mathcal{T}_k)/\sum_{k=1}^{K} \gamma(\mathcal{T}_k)$.

## 6 Experiments

**Dataset.** Argoverse [10] is one of the largest publicly available motion forecasting datasets, which contains over 324K data sequence collected from complex urban driving scenarios. The training, validation, and test sets are taken from disjoint parts of the cities. Each sequence lasts for 5 seconds, containing the centroid locations of each tracked agent sampled at 10 Hz, in which one vehicle with relatively complex motion is marked as the prediction target. The objective is to predict its locations 3 seconds into the future, given an initial 2-second observation.

**Metrics.** We follow the Argoverse evaluation criteria under $K = 1$ and $K = 6$. Minimum Average Displacement Error ($\text{minADE}_K$) is the average L2 distance error of the *best* predicted trajectory. Minimum Final Displacement Error ($\text{minFDE}_K$) is the L2 distance error of the *best* predicted trajectory at the final timestamp. Miss Rate ($\text{MR}_K$) is the ratio of scenarios that none of $K$ predicted trajectories has less than 2 meters L2 final displacement error. For multimodal prediction, the probability-based metrics p-$\text{minADE}_K$ and p-$\text{minFDE}_K$ are calculated by adding $-log(p)$ to $\text{minADE}_K$ and $\text{minFDE}_K$, where $p$ corresponds to the probability of the *best* predicted trajectory. In the Argoverse benchmark, *best* refers to the predicted trajectory with the minimum endpoint error.

**Implementation Details.** Our implementation is detailed in the supplementary material. Among the state-of-the-art methods, only LaneGCN [34] is open-source. So we use its official implementation and Argoverse baselines [10] for additional tests about trajectory feasibility and imperfect tracking.

| Method | K=1 | | | K=6 | | | | | Infeasibility (%) |
|---|---|---|---|---|---|---|---|---|---|
| | minADE | minFDE | MR (%) | minADE | minFDE | p-minADE | p-minFDE | MR (%) | |
| Argo-CV | 3.53 | 7.89 | 83.48 | 3.39 | 7.57 | 5.18 | 9.36 | 81.68 | 0.00 |
| Argo-LSTM+map | 2.96 | 6.81 | 81.22 | 2.34 | 5.44 | 4.14 | 7.23 | 69.16 | 43.53 |
| Argo-NN+map | 3.65 | 8.12 | 83.55 | 2.08 | 4.03 | 3.87 | 5.82 | 58.21 | 86.39 |
| LaneGCN [34] | 1.71 | **3.78** | 59.05 | **0.87** | **1.36** | **2.66** | 3.16 | 16.34 | 16.52 |
| Alibaba-ADLab | 1.97 | 4.35 | 63.76 | 0.92 | 1.48 | 2.67 | 3.23 | 15.86 | – |
| TNT [25] | 1.78 | 3.91 | 59.72 | 0.94 | 1.54 | 2.73 | 3.33 | 13.28 | – |
| Jean [21] | 1.74 | 4.24 | 68.56 | 1.00 | 1.42 | 2.79 | 3.21 | 13.08 | – |
| Poly | **1.70** | 3.82 | 58.80 | 0.87 | 1.47 | 2.67 | 3.26 | 12.02 | – |
| PRIME (Ours) | 1.91 | 3.82 | **58.67** | 1.22 | 1.56 | 2.71 | **3.05** | **11.50** | **0.00** |

Table 1: Comparison with the Argoverse baselines and the state-of-the-art methods on the Argoverse test set. All metrics are lower the better and Miss Rate (MR, K=6) is the key metric.

## 6.1 Comparison with State-of-the-art

We compare our proposed PRIME against the Argoverse baselines [10] (CV, LSTM+map, NN+map), the top-3 methods in the Argoverse Motion Forecasting Competition 2020 (Jean [21], Poly, Alibaba-ADLab), and the recently published state-of-the-art, LaneGCN [34] and TNT [25]. The performance comparison under Argoverse test set is shown in Table 1. It could be noted that PRIME outperforms all other methods on Miss Rate ($K = 6$), which is the official ranking metric in Argoverse Competition 2020. It reflects that our method produces accurate multimodal predictions consistently in diverse scenarios. We also achieve the best on the probability-based metric p-$minFDE_6$, which would be highly beneficial to weigh between multiple predictions in making decisions and motion plans. From the methods with public details, including LaneGCN [34], TNT [25], and Jean [21], we can find they all perform the learning-based paradigm that utilizes neural networks to model traffic entities and generates future trajectories, while PRIME is the only one that integrates a model-based motion generator into prediction learning. Notably, due to the lack of more detailed on-road information in the Argoverse dataset, such as vehicle types, bounding box, static obstacles, etc., the quantitative result is achieved by imposing general constraints on the model-based generator. This indicates there exists more space to improve when deploying our framework in a real autonomous driving system. Furthermore, handling environmental and dynamic constraints in an interpretable model-based manner and generating trajectories with continuous state information is significant for real-world deployment, which could not be reflected from the evaluation metrics.

## 6.2 Ablation Studies

We ablate the F2F module and Frenét representation (denoted by SD) from the complete evaluation network to study their impacts. Table 2 reports the results on the Argoverse validation set. With P2T, P2A, and A2A attention modules capturing the basic interactions between map and agents, the base model performs at the same level with TNT ($MR_6 = 9\%$ reported in [25]), indicat-

| Modules | p-$minADE_6$ | p-$minFDE_6$ | $MR_6$(%) | # Params |
|---|---|---|---|---|
| Base | 2.33 | 2.63 | 8.52 | 0.69 M |
| Base+F2F | 2.31 | 2.61 | 8.23 | 0.72 M |
| Base+SD | 2.29 | 2.58 | 7.81 | 0.99 M |
| Base+F2F+SD | **2.29** | **2.57** | **7.51** | 1.02 M |

Table 2: Ablation studies on the Argoverse validation set.

ing that these modules are effective in capturing agent-map interactions. As for the Frenét representation providing the local spatial relationship and the F2F module fusing all feasible trajectories to get a global understanding of the reachable space, they both promote the performance. By comparison, the inclusion of Frenét representation is more effective. Additionally, the complete network makes the best performance with only 1.02M parameters, which indicates that separating the function of trajectory generation would reduce the learning burden while achieving high performance.

## 6.3 Trajectory Feasibility

As a typical non-holonomic motion system, vehicles are constrained by inherent kinematic characteristics. So we investigate the ratio of infeasible trajectories produced by prediction models. Since the high-order states (velocity, acceleration, or turning rate) cannot be estimated accurately from discrete locations predicted by common learning-based models, we evaluate the trajectory feasibility only using curvature. By interpolating the predicted positions with pairwise cubic splines, we get the curvature at each point. A trajectory is labeled as infeasible if the curvature $\kappa > 1/3$ (i.e., the minimum turning radius is 3 meters) at any of its points. The ratio of infeasible trajectories is shown in the last column of Table 1. Except for the physical baseline Argo-CV (Constant Velocity), the others, as representatives of the unconstrained learning-based models, have at least $16.5\%$ infeasible predictions. Although we only use curvature for judgment and set a fairly conservative threshold (the minimal turning radius for a regular sedan is around $4.5 \sim 6.0$ meters), the infeasible predictions still take up a considerable proportion, which would cause redundant burdens for SDVs to make decisions and plans. By contrast, the model-based generator in our framework can handle any kinematic and environmental constraints, thereby ensuring trajectory feasibility.

## 6.4 Imperfect Tracking

While most motion forecasting datasets provide tracking results of a certain duration for prediction targets, a self-driving vehicle would inevitably encounter real-world situations where the target is lost in some timestamps or not tracked long enough yet. Then the prediction model is required to robustly handle imperfect tracks rather than being restricted to fixed-duration tracking inputs. To let the models (ours, LaneGCN, and NN+map baseline) be aware of imperfect tracks, we re-trained them by randomly dropping out tracked locations. For processing such inputs while keeping network structures, we pad the locations of dropped timestamps with the nearest tracked location and add a dimension of the binary mask to denote the padded location. The drop rate is randomly sampled from $0 \sim 0.6$ for each data

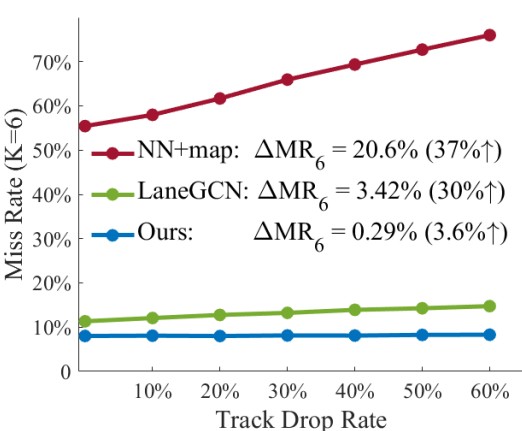

Figure 4: Comparison of prediction robustness under imperfect tracking.

sequence in training but fixed in testing. The drop rate is pointwise applied, i.e., 0.6 drop rate may drop more or less than $60\%$ of locations on a track. The last timestamp is always kept to ensure the prediction target could be detected at inference. Fig. 4 shows how the miss rate varies with track drop rate, we observe that our model performs stably, with only $3.6\%$ relative increase on $\text{MR}_6$, while the relative increase is around $30\% \sim 40\%$ for the others. The result indicates that the learning-based prediction models rely on long-term tracked results to regress trajectories, while our framework design relieves that to a certain extent, thereby improving the prediction robustness.

## 7 Conclusion

We present the prediction framework PRIME that learns to predict vehicle trajectories with model-based planning. PRIME guarantees the trajectory feasibility by exploiting a model-based generator to produce future trajectories under explicit constraints. It makes accurate trajectory predictions by employing a learning-based evaluator to capture implicit interactions in scene context and select future trajectories by scoring. With the novel framework design, PRIME outperforms the state-of-the-art in prediction accuracy, feasibility, and robustness. Moreover, the advantages of reasonably regularizing trajectory space, predicting trajectories with continuous state, and the compatibility with on-road information would set our framework highly useful in real system deployment.

## Acknowledgments

This research work was supported in part by The Hong Kong University of Science and Technology under the project "LDSERF-Autonomy Through Learning".

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
