# OpenReview forum: "Learning to Predict Vehicle Trajectories with Model-based Planning"
_robot-learning.org/CoRL/2021/Conference — CoRL2021 Poster_

### Official Review · Reviewer_PMNK · 2021-07-20

**Originality:** Good
**Technical Quality:** Good
**Clarity Of Presentation:** Good
**Impact:** 2

**Recommendation:**

Weak Accept: I recommend accepting the paper, but will not argue for my recommendation if the majority of other reviewers have a different opinion.

**Summary:**

This paper proposes a novel prediction framework called PRIME that learns to predict vehicle trajectories with a two-part approach with a model-based generator and a learning-based evaluator. This two-part system allows PRIME to generate accurate and feasibility-guaranteed predictions.

**Issues:**

The paper is well written and tackles a well-studied but
still relevant problem. The authors provide a good
literature review and clearly explain their architecture,
presenting some experiments to support their claim.  I believe it is suitable for CoRL.

**Reviewer Expertise:**

Fair: Some knowledge of the area

**Strengths And Weaknesses:**

1) I would suggest that the authors consider adding a discussion of the limitations of the approach to the end of the paper.

Minor comments

1) Figure 4 is small and difficult to read
2)Figure 3 caption is too long and should be reworked

**Summary Of Recommendation:**

The paper is well written and tackles a  relevant problem. The authors give a thorough description of their approach as well as experiments validating their claims. The author's two-part approach to trajectory prediction is novel and seems promising.

However, a more robust experiment where more detailed on-road information would give a better picture of how much the PRIME improves feasibility when compared to the other baselines.

Also important is the need of a more
developed discussion around the limits of the approach.

---

> ### Author Response · Authors · 2021-08-30
> **Author Response to Reviewer PMNK**
>
> Dear Reviewer PMNK,
>
> We sincerely thank your positive feedback and efforts for improving our manuscript.
> In the following, we have addressed each comment in detail.
>
> ---
>
> **Q1. I would suggest that the authors consider adding a discussion of the limitations of the approach to the end of the paper.**
>
> **A1.**
> For satisfying the requirement of page number limit, we add the Section D - Limitation and Future Work in the supplementary material, as detailed below:
>
> The framework could be further improved from the following aspects.
> We use some fixed parameters in the model-based generator, but better strategies can be applied when the required information is given.
> Firstly, the distance thresholds in the path search phase could be adjusted according to the target vehicle's state, and the resulted paths could be pruned by considering the lane connectivity given by curbs, fences, etc.
> Secondly, the trajectory generation phase could be refined by adjusting the lateral and longitudinal sampling boundaries based on the speed limits and in-place lane width, and adopting different sampling densities according to the target's impact (e.g., distance) on the autonomous vehicle.
> All these adjustments would contribute to alleviating the computational cost in the model-based generator.
> For the learning-based evaluator, separating the function of trajectory generation enables it to achieve good performance using a lightweight network with 1.02M parameters, which also leaves space for optimizing the network structure.
> We plan to extend scene encoding from reachable paths to a lane graph (similar to VectorNet [33] or LaneGCN [34]), where a complete context encoding is expected to bring performance improvement.
>
> ---
>
> **Q2. Figure 4 is small and difficult to read. Figure 3 caption is too long and should be reworked.**
>
> **A2.**
> In the revised manuscript, we streamline the caption of Figure 3 and scale up Figure 4 for better readability.

---

> ### Comment · Reviewer_PMNK · 2021-09-04
> **Final Recommendation**
>
> After reading the author's response, I'd like to maintain my score which is weak accept

---

### Official Review · Reviewer_fWJ2 · 2021-07-22

**Originality:** Very Good
**Technical Quality:** Very Good
**Clarity Of Presentation:** Very Good
**Impact:** 4

**Recommendation:**

Strong Accept: I recommend accepting the paper and will argue for my recommendation even if other reviewers hold a different opinion.

**Summary:**

This paper proposes a technique to generate robust and feasible vehicle trajectory predictions. A model-based motion planner is designed to search reachable reference paths, around which the candidates of trajectory predictions are sampled, guaranteed to be compliant with kinematic constraints and road structures. The trajectories will be scored by a learning-based evaluator, which takes map structure and agent interaction into consideration.

**Issues:**

I am curious about the forward inference time.

**Reviewer Expertise:**

Very good: Comprehensive knowledge of the area

**Strengths And Weaknesses:**

Strengths:
1) The paper is well-written and easy to follow.
2) The way to utilize map information is interpretable, and results are guaranteed to be feasible.
3) Most model-based methods fail to generate diverse trajectories in interactive scenes, while learning-based methods suffer from trajectory quality. This paper combines the advantages of both methods and performs well in the argoverse competition.

Weaknesses:
1) The novelty of this paper is the combination of a model-based planner and a learning-based evaluator. However, the planner itself is widely used in industry, and the evaluator follows LaneGCN’s structure. I am not sure about the inference time for this proposed method.

**Summary Of Recommendation:**

I would recommend this paper because it proposes a new method to make trajectory generators more practical. The experimental results are convincing.

---

> ### Author Response · Authors · 2021-08-30
> **Author Response to Reviewer fWJ2**
>
> Dear Reviewer fWJ2,
>
>
> We sincerely thank your constructive and insightful analysis, and appreciation of our method's novelty, performance, and presentation clarity.
> In the following, we have addressed each comment in detail.
>
> ---
>
> **Q1. The novelty of this paper is the combination of a model-based planner and a learning-based evaluator. However, the planner itself is widely used in industry, and the evaluator follows LaneGCN’s structure.**
>
> **A1.**
> We appreciate the reviewer's recognization of this novel framework for trajectory prediction.
> The model-based generator is based on the Frenet planner [17], commonly used to plan the autonomous vehicle but adapted for future trajectory generation in our prediction framework.
> While for the learning-based evaluator, we would like to explain that PRIME differs from LaneGCN [34] from design objective to network structure.
> Conceptually, in view that the model-based generator can provide trajectories with sufficient coverage to the future space, our evaluator is designed for selecting trajectories by scoring, rather than generating trajectories as most prediction (including LaneGCN) works did.
> Owing to separating the functionality of trajectory generation from the network, our evaluator associates the observed tracks from the history time domain with the generated trajectories from the future time domain, without the need for a multi-modal prediction header. Therefore, except for adopting the attention mechanism in interaction learning, our proposed evaluator network has critical differences compare with LaneGCN.
>
> ---
>
> **Q2. About the inference time for this proposed method.**
>
> **A2.**
> We provide the runtime analysis in Section C of the supplementary material.
> The model-based generator is implemented with Python on Intel i7-7820X. The generation of a single trajectory with a single thread spends $0.1\sim0.2$ ms on average.
> As each trajectory sample is produced independently, the model-based trajectory generator could be highly parallelized. Each prediction target in the dataset finally obtains 484 feasible trajectories on average, while the required number of trajectories would be reduced when the detailed road information (e.g., speed limits, lane width, vehicle type) could be accessed and used as constraints.
> For the learning-based evaluator, it is implemented by a lightweight network with $8\sim12$ ms inference time on NVIDIA 2080TI.
> By statistic, our implementation of the PRIME framework gets $18\pm4$ ms inference time for all scenarios in the Argoverse dataset, which could well satisfy the real-time requirements for autonomous driving.

---

### Official Review · Reviewer_SJ8P · 2021-07-26

**Originality:** Fair
**Technical Quality:** Good
**Clarity Of Presentation:** Good
**Impact:** 3

**Recommendation:**

Weak Accept: I recommend accepting the paper, but will not argue for my recommendation if the majority of other reviewers have a different opinion.

**Summary:**

The authors propose a prediction model that ranks different plans generated conditioned on reachable paths in the road network. A planner submodule is used to plan along the path, and an evaluator is used to rank the trajectories after encoding them. The authors detail the structure of the evaluator and choice of planner submodule, post-processing selection, as they obtain favorable results on Argoverse.

**Issues:**

* How does the approach depend on the quality and diversity of the proposals made under the assumption of limited number of samples (MoN, N~5..7)? Do more suggested plans hurt performance, does the performance depend on good and “representative” action set? (the main problem is that it’s hard to generalize a models and plans to cover edge cases, which is why people look at model-free trajectory generation)
* Why do we want planning-based predictions? Are we learning feasible or rewarding trajectories, and if so, why the assumption on a reward/planning structure? How does this compare to other approaches such as IRL / reward shaping (“Maximum Entropy Deep Inverse Reinforcement Learning“, “Social behavior for autonomous vehicles”), learned planning for  corrections of predictions (“Differentiable Logic Layer for Rule Guided Trajectory Prediction”), and injection into the discriminator (“Vehicle Trajectory Prediction Using Generative Adversarial Network With Temporal Logic Syntax Tree Features“) beyond citing, as the heavier machinery is in writing planners, seems like testing the other approaches for a comparison would’ve been easy and worthwhile. Would be also good to cite “What the Constant Velocity Model Can Teach Us About Pedestrian Motion Prediction“, as it makes a relevant claim -- learning-based generators are still quite sub-optimal given the motion statistics, to the point that simplified models (such as path following) can outperform them.
* While it’s interesting to ablate the local/global coordinates representation and the F2F attention, the interesting ablations would be to compare the evaluator and planner contributions, since individually they have appeared in several works. Moreover, while adhering to lanes is important, this has been used in predictors before, so would be interesting to compare the specific representation for lane following vs. other approaches.
* Where are the cases where planning is not adhered to and how bad are such mistakes? The main reason for model-free prediction is to handle rare events, but these are not always captured by path following.
* In cases where multiple trajectories are generated for the same path, or nearby examples -- how do they differ? What do they capture? Would be good to hear/see examples, in addition to failure cases shown in the supp. mat.


**Reviewer Expertise:**

Good: General knowledge of the area

**Strengths And Weaknesses:**

Strengths: Results  on Argoverse seem good, and some ablations show the exploration of design tradeoffs. Model-based trajectories can improve prediction at long horizons for majority of the driving scenarios, and can leverage the rich literature on planners, including different goals, rewards and constraints. I liked the exploration of tracking impact as it’s often neglected, and the use of local/global coordinates, which make sense and has been also shown in the recent CVPR WAD workshop discussions.

Weaknesses: The paper’s main claim to fame (prediction via planning) and the main structure (i.e use of ranking) has been significantly explored before. While the specific approach is new, the readers would benefit from a more balanced view that targets the (much needed) set of open questions in the field that the paper touches on, such as exploration of the impact of the choices for planner, evaluator, lane handling, planner sample parameters (the path selected, any other parameters that prevent impoverished prediction set?).


**Summary Of Recommendation:**

While the approach is not completely novel, and reuses elements from many existing works, it is overall a new combination, is presented clearly, and achieved good result on a valid benchmark. Would be good to see a better presentation with respect to other works that do planning for prediction, or generate-then-evaluate, and to explore the tradeoffs that drove the error down for this system.

---

> ### Author Response · Authors · 2021-08-30
> **Author Response to Reviewer SJ8P (1/4)**
>
> Dear Reviewer SJ8P,
>
> We sincerely thank your insightful comments, detailed feedback for clarity, and appreciation of our exploration of prediction robustness under imperfect tracking and utilization of local and global representations.
> In the following, we have addressed each comment in detail.
>
> ---
>
> **Q1. How does the approach depend on the quality and diversity of the proposals made under the assumption of limited number of samples (MoN, N~5..7)? Do more suggested plans hurt performance, does the performance depend on good and “representative” action set?**
>
> **A1.**
> As the learning-based evaluation in our framework only takes charge of selecting trajectories, rather than regressing any positions or displacement, the prediction trajectory $\mathcal{T}_{tar}$, that is selected from the trajectory set $\mathcal{T}$, is directly affected by the model-based generator.
> The provided set of future trajectories $\mathcal{T}$ need to sufficiently cover the future trajectory space.
> The impact of its quality and diversity is demonstrated in the failure cases caused by inaccurate state estimated from sequences of centroid positions, in Figure 9(a) of the supplementary material, where the ground truth locates outside the generated trajectory space.
>
> We have experimentally validated that the performance gains improvement from more trajectory proposals. Considering the trade-off between sampling density and computational cost, as well as the little difference in performance when using more dense trajectories, we settle down to the current set of parameters.
> Additionally, we do not specifically design some "representative" actions but exploit the model-based generator to produce trajectories set that covers feasible driving maneuvers.
>
> ---
>
> **Q2. Why do we want planning-based predictions?**
>
> **A2.**
> Compared with the mainstream learning-based prediction, our work has validated that leveraging a model-based generator to produce future trajectories in prediction would bring benefits in:
> - Maintaining multimodal distributions by generating trajectories on diverse reachable paths in an interpretable approach,
> - Ensuring trajectory feasibility by imposing real-time constraints,
> - Mitigating the high reliance on long-term tracking, and
> - Producing trajectories with continuous information rather than just discrete locations.
>
> By making the advantages of the model-based generator and the learning-based evaluator, the proposed framework of "Learning to Predict with Model-based Planning" is capable of handling complex agent-map interactions while fulfilling environmental and kinematic constraints in vehicle trajectory prediction.
>
> ---
>
> **Q3. Are we learning feasible or rewarding trajectories, and if so, why the assumption on a reward/planning structure?**
>
> **A3.**
> It should be noted that the intersection between our framework and planning methods only lies in the trajectory generation phase, while our framework excludes any cost / reward-based evaluation that is commonly used for optimal trajectory selection in planning.
> Specifically, for the trajectory generation phase, there is no learning but only relying on the model-based generator to sample trajectories $\mathcal{T}$ that are compliant with given constraints.
> For the evaluator, rather than learning feasible or rewarding trajectories, the learning objective is to model the implicit map-agent interaction and select the probable future trajectories $\mathcal{T}_{tar}$ from the given feasible trajectory set $\mathcal{T}$.

---

> > ### Author Response · Authors · 2021-08-30
> > **Author Response to Reviewer SJ8P (2/4)**
> >
> > **Q4. How does this compare to other approaches such as IRL / reward shaping (“Maximum Entropy Deep Inverse Reinforcement Learning“, “Social behavior for autonomous vehicles”), learned planning for corrections of predictions (“Differentiable Logic Layer for Rule Guided Trajectory Prediction”), and injection into the discriminator (“Vehicle Trajectory Prediction Using Generative Adversarial Network With Temporal Logic Syntax Tree Features“) beyond citing, as the heavier machinery is in writing planners, seems like testing the other approaches for a comparison would’ve been easy and worthwhile.**
> >
> > **A4.**
> > Not limited to the mentioned works, the existing prediction approaches are either incapable of providing feasibility-guaranteed trajectory prediction or implemented with handcrafted rules/models to process multi-agent interactions.
> > PRIME is the first to incorporate an interpretable motion planner in prediction learning to the best of our knowledge.
> > The combination of model-based generator and learning-based evaluator makes it capable of dealing with various hard constraints (which is hard to realize with fully learning-based frameworks) and modeling agent-map interactions in a data-driven manner (which is difficult to tackle with model-based approaches).
> >
> > The most critical contribution is the novel framework design, enabling state-of-the-art prediction performance, trajectory feasibility guarantee, robustness under imperfect tracking, and better interpretability.
> > In contrast, the adaption in the generator and the network design for the evaluator is secondary.
> > We believe it is convincing by the extensive comparison with the leading methods in the highly competitive Argoverse Competition, so we have not added more comparisons.
> > We thank the reviewer for showing these works and add the related ones to the reference list.
> >
> > ---
> >
> > **Q5. Would be also good to cite “What the Constant Velocity Model Can Teach Us About Pedestrian Motion Prediction“, as it makes a relevant claim -- learning-based generators are still quite sub-optimal given the motion statistics, to the point that simplified models (such as path following) can outperform them.**
> >
> > **A5.**
> > The mentioned work does reveal some interesting facts in pedestrian trajectory prediction. However, there exist significant differences between pedestrian prediction and vehicle prediction.
> > - Motion state: compared with the on-road vehicles that move with a large range of speed, pedestrians mostly move at a relatively fixed speed.
> > - Environmental and kinematic constraints: compared with vehicles that operate under their inherent kinematic constraints while in compliance with the structured road environment, pedestrians are more like omnidirectional motion systems that commonly move in open spaces (sidewalk, square, and indoor environments, etc.)
> >
> > As a result, the performance between the Constant Velocity Model and the learning-based methods in pedestrian trajectory prediction may differ not much, but the situation is completely different on vehicle trajectory prediction.
> > As indicated by the leading methods on Argoverse, all of them are learning-based, while PRIME is the only one that incorporates a model-based generator into prediction learning.
> > By contrast, the Constant Velocity Model in the Argoverse official baselines (Argo-CV in Table 1) performs much worse in all evaluation metrics.
> > So the claim made by the mentioned paper does not hold for the vehicle trajectory prediction task.

---

> > > ### Author Response · Authors · 2021-08-30
> > > **Author Response to Reviewer SJ8P (3/4)**
> > >
> > > **Q6. While it’s interesting to ablate the local/global coordinates representation and the F2F attention, the interesting ablations would be to compare the evaluator and planner contributions, since individually they have appeared in several works. Moreover, while adhering to lanes is important, this has been used in predictors before, so would be interesting to compare the specific representation for lane following vs. other approaches.**
> > >
> > > **A6.**
> > > It is necessary to claim the critical differences between our generator and evaluator with the existing works.
> > > The trajectory generator is adapted from Frenet Planner [17], but the objectives are quite different.
> > > Unlike planning autonomous vehicles where a reference path would be given, prediction cannot know which path the predicted targets would move towards.
> > > It requires searching paths that are possibly reachable before generating trajectories to cover the future prediction space.
> > > The optimal trajectory strategy in Frenet Planner cannot be used for directly producing a trajectory as a prediction result, as the cost function for selection is related to a reference path, while prediction may associate with multiple paths.
> > > Therefore, it is impossible to evaluate the generator without a learning-based evaluator.
> > >
> > > To the best of our knowledge, using an evaluator to rank a varying number of future trajectories in prediction is initiated in this work, which is also recognized by Reviewer Q68c:
> > > > The method is quite different from a more mainstream approach of using deep NN to generate trajectories but instead learning an evaluator. This is important and can inspire many more avenues in this direction.
> > >
> > > Similarly, the characteristic of separating the function of trajectory generation from the network makes it impossible to ablate the model-based generator from the framework.
> > >
> > > Lastly, the implemented lane-following predictor in [3] makes each predicted vehicle simply follow the lane and maintain its distance to the right bound, which is infeasible in most urban driving scenarios.
> > > By comparison, our prediction framework performs well in complex urban scenarios by considering all the reachable paths to produce trajectories with lane-keeping and various lane-changing behaviors, as illustrated in each qualitative example.
> > >
> > > ---
> > >
> > > **Q7. Where are the cases where planning is not adhered to and how bad are such mistakes?**
> > >
> > > **A7.**
> > > We did consider whether there exists some actual trajectory that the trajectory generator cannot cover. The only case is that a vehicle performs multiple actions within the prediction time horizon (i.e., around 3s), such as changing lanes back and forth, frequently accelerating and decelerating. Although it rarely happens, we do not believe it is a shortcoming for the trajectory generator in our framework.
> > > This is because the planning process of both human drivers and motion planners performs in a receding horizon manner, i.e., planning a 3-second trajectory at the current frame and again planning the subsequent 3-second trajectory at the next frame. Therefore, each actual trajectory (recorded as the ground truth to judge if the prediction is good or not) results from multi-cycle planning rather than a single cycle.
> > > Similarly, the prediction process operates in the same manner as planning.
> > > So the trajectory generator in our prediction framework is designed to cover the target's future trajectory that is possible at the current frame. In this way, it may not generate a trajectory similar to the rare ground truth trajectories that contain multiple actions but is still highly useful for downstream planning if the prediction results converge frame by frame.
> > >
> > > ---
> > >
> > > **Q8. The main reason for model-free prediction is to handle rare events, but these are not always captured by path following.**
> > >
> > > **A8.**
> > > Firstly, we would like to explain that our trajectory generator is not limited to lane following but sampling towards all the paths that are possibly reachable. Therefore, as each qualitative result illustrates, the predicted driving behaviors include lane-keeping and various lane-changing behaviors.
> > >
> > > Although the model-free methods are expected to handle rare events, the resulted constraint-free predictions contain some infeasible trajectories and bring massive uncertainty in the predicted future states.
> > > As demonstrated by the comparisons in Figure 6~7, the extra burden of infeasible trajectories would eventually impose on the downstream planning and even incurs the "freezing robot problem" [11].
> > > While for those rare behaviors that would never be produced by our trajectory generator, such as driving on the reverse lanes or at a much higher speed than limits, they are not hard to be recognized.
> > > Therefore, compared with adopting a constraint-free predictor, it would be more effective to handle rare behaviors with an "abnormal detector" and infer future trajectories with a feasibility-guaranteed predictor.

---

> > > > ### Author Response · Authors · 2021-08-30
> > > > **Author Response to Reviewer SJ8P (4/4)**
> > > >
> > > > **Q9. In cases where multiple trajectories are generated for the same path, or nearby examples -- how do they differ? What do they capture? Would be good to hear/see examples, in addition to failure cases shown in the supp. mat.**
> > > >
> > > > **A9.**
> > > > For the trajectories that are generated for the same path (a sequence of lane centerlines), they differ in longitudinal movement $s(t)$ and lateral movement $d(t)$.
> > > > Since $s(t)$ and $d(t)$ are generated independently by connenect the current state as the fixed start state with different end state at the prediction horizon $T_F$,
> > > > i.e., the target longitudinal velocity $\dot{s}(T_F)$ and the target lateral offset $d(T_F)$.
> > > >
> > > > Each resulted $s(t)$ denotes a velocity profile and each $d(t)$ indicates a lateral behavior.
> > > > Therefore, each full trajectory $\vec{x}(s(t), d(t))$, formed by a combination in $\mathcal{T}_{lon} \times \mathcal{T}_{lat}$, covers a specific driving maneuver.
> > > > The future encoder would capture the difference between trajectories, which is implemented by a temporal convolutional layer followed by an LSTM layer.
> > > >
> > > > We would like to refer the reviewer to the trajectory sets (blue ones) in Figure 5~9 of the supplementary material, where each trajectory would explain how to move towards / away from the centerline of each corresponding reference path, and they all together cover various driving maneuvers.

---

### Official Review · Reviewer_Q68c · 2021-07-29

**Originality:** Very Good
**Technical Quality:** Excellent
**Clarity Of Presentation:** Excellent
**Impact:** 4

**Recommendation:**

Strong Accept: I recommend accepting the paper and will argue for my recommendation even if other reviewers hold a different opinion.

**Summary:**

This paper aims to generate trajectories that abide by both kinematic and contextual constraints. The kinematic constraints include motion planning feasibility. The contextual constraints include road compliance, following traffic lights, following the speed limits. The key idea of this paper is to (1) generate a lot of plans and trajectories (2) train an evaluator to give the scores each trajectory.

The plans and trajectories are generated by doing depth first search in HD map for plans, and then use the dynamics to figure out the trajectories.

The evaluator is trained with the objective that regresses the predicted score to the labeled score using a cross entropy loss. The labeled score is computed by how far they are from the ground truth trajectory. The model takes both states, paths, and trajectories from the previous step, and pass them through a bunch of attention layers and output the score.

The authors show low p-ADE and miss rate meaning that it can cover the ground truth trajectory within 2 meters within one of the 6 predicted trajectories with 89% accuracy on Argoverse. The authors show that it is robust when tracking drops.

**Issues:**

What are the failure modes of missing trajectories in the 11%? Can you provide common example scenarios?

How realistic is the tracking dropping? The experimental settings seem a bit constructed. Were there some references to that this is a common issue in practice?

**Reviewer Expertise:**

Good: General knowledge of the area

**Strengths And Weaknesses:**

The paper is well-written and clear.
The method is quite different from a more mainstream approach of using deep NN to generate trajectories but instead learning an evaluator. This is important and can inspire many more avenues in this direction.
The results demonstrate the improvement over the previous state-of-the-arts. The paper shows the robustness when tracking is dropped.

**Summary Of Recommendation:**

The paper proposes an interesting idea of combining planning and learning in motion forecasting. The results surpassed other SOTA. While the paper does a great job considering robustness to tracking, I wonder if there is a reference of the actual problem in tracking.

---

> ### Author Response · Authors · 2021-08-30
> **Author Response to Reviewer Q68c**
>
> Dear Reviewer Q68C,
>
> We sincerely thank your insightful and comprehensive summary of our work. We also appreciate your favorable impression regarding its novelty, performance, and presentation clarity.
> In the following, we have addressed each comment in detail.
>
> ---
>
> **Q1. What are the failure modes of missing trajectories in the 11%? Can you provide common example scenarios?**
>
> **A1.**
> Since the ground-truth trajectories in the test set of the Argoverse Competition are not accessible (the performance on the test set is assessed on the Argoverse official server), we cannot recognize the failure modes in the test set. We analyzed the failure cases from the Argoverse evaluation set, in which PRIME gets a 7.51% Missing Rate (reported in Table II). The most common failure cases originate from inaccurate state estimation and high-speed scenarios, as shown in Figure 9 and analyzed in Section B.4 of the supplementary material.
>
> One type of common failure results from inaccurate state estimation for the prediction target. Although the sampling-based strategy in our generator could compensate for the inaccuracy to some extent, estimating the heading and velocity from the sequences with only centroid positions given by Argoverse would be intractable when serious data noise exists. While in the autonomous driving systems, the vehicle's bounding box given by detection provides geometry information in addition to discrete positions, which would enable more robust and accurate state estimation for prediction targets.
>
> Another common failure typically occurs in high-speed scenarios where the prediction target moves fast towards its forward open space. In this case, the 3-second future trajectory space would be much larger and naturally leads to a higher probability of missed predictions (minFDE$_6$>2m).
> Nonetheless, it could be observed that our predictions, locating within a compact feasible trajectory space, accurately capture the target's intention with an acceptable displacement error, which makes sense for the downstream decision-making and planning.
>
> ---
> **Q2. How realistic is the tracking dropping? The experimental settings seem a bit constructed. Were there some references to that this is a common issue in practice?**
>
> **A2.**
> Imperfect tracking is an inevitable issue that would happen when deploying autonomous vehicles in real-world situations.
> Whatever the specific tracking method is applied, imperfect tracking would be commonly raised by the short-term observation and occlusion in complex traffic scenarios and practical engineering issues.
> Specifically, the tracking sequence cannot be accumulated enough (2s in the Argoverse Benchmark) when the target vehicle appears within our sensing range for a short time. The state information under the timestamps when the target vehicle is occluded would also be absent in the tracking sequence.
> Furthermore, at the engineering level, the inference time fluctuation of the upstream detection and tracking often makes some frames dropped while it occurs randomly.
> Considering these, we validate the prediction robustness by randomly dropping the tracking sequence with different rates.
> This approach covers all the realistic cases of imperfect tracking.

---

> > ### Comment · Reviewer_Q68c · 2021-09-01
> > **Thank you**
> >
> > Thank you for answering my questions and concerns. After reading the author's response, I'd like to maintain my score which is strong accept.

---

### Meta-Review · Area_Chair_CeCH · 2021-08-14

**Recommendation:** Accept (Poster)
**Confidence:** 4

**Metareview:**

Summary: This work proposes a prediction mechanism that combines model-based path and trajectory planners with a learned ranking system. The proposed method is supported by results on the Argoverse Benchmark.

Clarity: All reviewers commented on the good composition of the paper. They seemed to agree that the paper was clearly and concisely written and overall effective in convincing the reader of the promise of the proposed approach.

Originality: Reviewers noted that while the paper adopts existing techniques and a popular framing of the problem, it does combine specific techniques in a novel way.

Quality: Reviewers were overall convinced that the results presented on the Argoverse Benchmark demonstrated a believable improvement over the state-of-the-art, however, also made suggestions on how to make the case even stronger.

Significance: Reviewers had mixed feedback on the impact of the paper. It seems like reviewers were mainly looking to the experimental results to convince them and noted some gaps and lingering questions.

Pros:
The paper proposes a compelling way to combine model-based and learning-based tools to solve an important problem in robotics. The paper also addresses salient details such as the effect of tracking error and ablation experiments.

Main opportunities for improvement:
The authors are encouraged to focus efforts on addressing the questions raised and do the suggested additional analyses the reviewers requested in the experimental results section. The goal of these additions should be to resolve any lingering questions about the effectiveness of the proposed approach.

Thank you for considering our feedback, and we look forward to seeing the updated paper.

============== Final Decision

All reviewers recommend acceptance.

I am not 100% certain if this should be a poster v an oral presentation. The case for oral is that two reviewers gave a strong accept with strong scores in all sub-categories. However, I ultimately recommended poster because sub-scores in impact and originality are mixed (in particular, 2 in impact from reviewer PMNK  and fair in originality from reviewer SJ8P).

---

> ### Author Response · Authors · 2021-08-30
> **Author Response to Area Chair CeCH**
>
> Dear Area Chair CeCH,
>
> We sincerely thank you for the comprehensive summary of reviews from all aspects and for pointing out the main improvement directions.
>
> We are grateful for receiving coincident positive recommendations for our work, including originality (R1,3,4), technical quality (R1,2,3,4), presentation clarity (R1,2,3,4), and experimental performance (R1,2,3,4).
> Furthermore, all the reviewers recognize the novel framework design that combines the advantages of a model-based generator and a learning-based evaluator and highly praise our effort on guaranteeing trajectory feasibility and considering imperfect tracking.
>
> As for the issues and concerns raised, we have carefully responded to each reviewer.  The main topics are listed as follows:
> -  The common scenarios of failure predictions (R1).
> -  The rationality in the experiments of dropping tracks (R1).
> -  The relationship between prediction performance and trajectory quality / diversity / number (R2).
> -  The motivation behind designing such a planning-based prediction (R2).
> -  The cases where planning cannot cover (R2).
> -  What is the difference between nearby trajectory samples, and what do these nearby trajectories capture (R2).
> -  The critical difference between our evaluator and LaneGCN (R3).
> -  The inference time for the whole framework (R3).
> -  The limitation of the proposed approach (R4).
>
> Correspondingly, we have revised the manuscript in the following parts:
> -  Additional examples and analysis for the common scenarios of failure predictions (Sec. B.4 & Fig. 9, supplementary material).
> -  Additional relevant references in the introduction and related work sections (Sec. 1 & Sec. 2)
> -  Limitations and future works (Sec. D, supplementary material)
> -  Other minor revisions (streamlined caption of Fig. 3, larger size for better readability of Fig.4, etc.)
>
> Please check the detailed reply in the respective review.

---

### Decision · Program_Chairs · 2021-09-13

**Decision:**

Accept (Poster)

**Comment:**

Summary: This work proposes a prediction mechanism that combines model-based path and trajectory planners with a learned ranking system. The proposed method is supported by results on the Argoverse Benchmark.

Clarity: All reviewers commented on the good composition of the paper. They seemed to agree that the paper was clearly and concisely written and overall effective in convincing the reader of the promise of the proposed approach.

Originality: Reviewers noted that while the paper adopts existing techniques and a popular framing of the problem, it does combine specific techniques in a novel way.

Quality: Reviewers were overall convinced that the results presented on the Argoverse Benchmark demonstrated a believable improvement over the state-of-the-art, however, also made suggestions on how to make the case even stronger.

Significance: Reviewers had mixed feedback on the impact of the paper. It seems like reviewers were mainly looking to the experimental results to convince them and noted some gaps and lingering questions.

Pros:
The paper proposes a compelling way to combine model-based and learning-based tools to solve an important problem in robotics. The paper also addresses salient details such as the effect of tracking error and ablation experiments.

Main opportunities for improvement:
The authors are encouraged to focus efforts on addressing the questions raised and do the suggested additional analyses the reviewers requested in the experimental results section. The goal of these additions should be to resolve any lingering questions about the effectiveness of the proposed approach.

Thank you for considering our feedback, and we look forward to seeing the updated paper.

============== Final Decision

All reviewers recommend acceptance.

I am not 100% certain if this should be a poster v an oral presentation. The case for oral is that two reviewers gave a strong accept with strong scores in all sub-categories. However, I ultimately recommended poster because sub-scores in impact and originality are mixed (in particular, 2 in impact from reviewer PMNK  and fair in originality from reviewer SJ8P).